# The impact of mutations on TP53 protein and MicroRNA expression in HNSCC: Novel insights for diagnostic and therapeutic strategies

Ashraf Attia Mahmoud[1], Mohd Firdaus Raih[2,3,4], Edison Eukun Sage [2], Qurashi M. Ali[5], Omnia H. Suliman[6], Sabah A. E. Ibrahim [1], Osama Mohamed[7], Samar Abdelrazeg[1], Sofia B. Mohamed [1,2]*

1 Department of Bioinformatics and Biostatistics, National University Biomedical Research Institute, National University-Sudan, Khartoum, Sudan, 2 Department of Applied Physics, Faculty of Science and Technology, Universiti Kebangsaan Malaysia, UKM Bangi, Selangor, Malaysia, 3 Institute of Systems Biology, Universiti Kebangsaan Malaysia, UKM Bangi, Selangor, Malaysia, 4 Bioinformatics and Molecular Simulations Research Group, Universiti Kebangsaan Malaysia, UKM Bangi, Selangor, Malaysia, 5 National University-Sudan, Khartoum, Sudan, 6 Department of Medicine and Surgery, Dubai Medical University, Dubai, United Arab Emirates, 7 Department of Molecular Biology, National University Biomedical Research Institute, National University-Sudan, Khartoum, Sudan

* sofiabashir2002@gmail.com

## Abstract

The tumor suppressor protein p53 (TP53) is frequently mutated in various types of human malignancies, including HNSCC, which affects tumor growth, prognosis, and treatment. Gaining insight into the impact of TP53 mutations in HNSCC is crucial for developing new diagnostic and therapeutic methods. In this study, we aimed to investigate the influence of mutations on the structure and functions of the TP53 protein and miRNA expression using computational analysis. The genomic data of patients with HNSCC were obtained from TCGA, and the impact of mutations on the TP53 gene was investigated using different bioinformatics tools. Results: The findings showed that the TP53 mutations increased TP53 expression levels in HNSCC and were associated with a poor prognosis. Furthermore, hsa-mir-133b expression was reduced in TP53-mutated samples, significantly affecting patient survival in HNSCC. Six mutations, including R273C, G105C, G266E, Q136H/P, and R280G, were identified as deleterious, carcinogenic, driver, highly conserved, and exposed. These mutations were located in the P53 domain, and PTM analysis revealed that R280G and R273C are at a methylation site, and R273C, Q136H/P, and R280G are located in the protein pocket. The docking research indicated that these mutations decreased the binding affinity for DNA, with R273C, R280G, G266E, and G105C displaying the most significant differences. The molecular dynamics analysis indicates that R280G, Q136H, and G105C mutations confer a gain of function by stabilizing the TP53-substrate complex. Conclusions: Based on the research findings, the mutations on TP53 were found to have an impact on protein and miRNA expression, development, survival, and progression of

**Data availability statement:** The data used in this study were obtained from publicly available repositories: -The Cancer Genome Atlas (TCGA) via the Genomic Data Commons (GDC) Data Portal: https://portal.gdc.cancer.gov/ -Data accession: TCGA-HNSC project (https://portal.gdc.cancer.gov/projects/TCGA-HNSC) -cBioPortal for Cancer Genomics: http://www.cbioportal.org-TCGAFirehoseLegacy: https://www.cbioportal.org/study/summary?id=hnsc_tcga -TCGA Nature 2015: https://www.cbioportal.org/study/summary?id=hnsc_tcga_pub -TCGA Pan Cancer Atlas: https://www.cbioportal.org/study/summary?id=hnsc_tcga_pan_can_atlas_2018 -UALCAN for Gene Expression Analysis: https://ualcan.path.uab.edu/ All datasets are publicly available and can be accessed using the links provided. No new datasets were generated for this study.

**Funding:** • [Universiti Kebangsaan Malaysia] – [DIP-2023-XX] • [National University-Sudan] – [NUSU, 2022,007].

**Competing interests:** The authors have declared that no competing interests exist.

HNSCC patients, and has-mir-133b could be a promising novel biomarker for monitoring the progression of HNSCC. It was discovered that G105C and Q136H/P, as novel mutations, affect the function and structure of proteins causing HNSCC, which indicates that they could be interesting subjects for further investigation, diagnostics, and therapeutic strategies. Furthermore, the precise positioning of R280G and R273C within the methylation site and Q136H/P in the binding site has been documented for the first time. Moreover, the G105C, Q136H, and R280G mutations that stabilized TP53 structure and altered its interaction dynamics with substrates may serve as novel potential diagnostic biomarkers in cancer, guiding patient stratification and personalized treatment strategies. The molecular dynamics analysis provides insights into how specific TP53 mutations impact protein structure, stability, and function upon substrate binding, highlighting their role in cancer biology and potential implications for therapeutic interventions. This paper provides a novel understanding of the mechanisms by which these mutations contribute to the development of cancer.

## 1. Introduction

Head and neck squamous cell carcinoma (HNSCC) is a heterogeneous group of cancers arising from the epithelial lining of the upper aerodigestive tract, including the oral cavity, pharynx, and larynx. HNSCC accounts for a significant portion of cancer-related morbidity and mortality worldwide, posing a substantial public health burden [1]. Despite the progress made in treatment approaches like surgery, radiation therapy, and chemotherapy, the prospects for patients still remain uncertain, underscoring the urgent need for improved understanding of the molecular mechanisms driving HNSCC tumorigenesis and progression [2]. Among the multitude of genetic alterations implicated in HNSCC pathogenesis, mutations within the tumor protein p53 (TP53) gene stand out as pivotal contributors to tumorigenesis and disease progression [3]. The tumor suppressor gene TP53, located on chromosome 17p13.1 and often referred to as the "guardian of the genome," plays a critical role in maintaining genomic integrity and orchestrating cellular responses to various stress signals, including DNA damage, hypoxia, and oncogenic stress. In cancer biology, TP53 mutations disrupt these essential functions, leading to genomic instability and unchecked cell proliferation, which are hallmarks of tumorigenesis [4]. The TP53 gene is frequently mutated in HNSCC, with alterations ranging from point mutations to large deletions, resulting in the loss of wild-type p53 function and the acquisition of oncogenic properties. By explaining TP53's tumor-suppressing mechanisms, we can better underscore its importance in HNSCC and the implications of its mutation for cancer progression. Many studies have demonstrated that manipulating miRNA expression might enhance the efficacy of mutant p53 proteins in several types of cancer, such as breast cancer, non-small cell lung carcinoma (NSCLC), and HNSCC [4]. The microRNAs, which are short non-coding RNAs, act as post-transcriptional regulators of gene expression. A significant number of miRNAs exhibit evolutionary conservation across diverse taxa, suggesting their vital involvement in fundamental

biological processes. The miRNAs control numerous cellular activities, including proliferation, differentiation, and cell death. Many miRNAs located in certain regions of the genome are either lost or overexpressed in human tumors. Studies have shown that p53, which is the gene most commonly altered in human cancer, controls the expression of miRNA. The TP53 gene predominantly undergoes missense mutations, resulting in the synthesis of a mutant p53 protein that retains its full length. This is in contrast to most tumor suppressor genes, which are usually inactivated by biallelic deletion or truncation mutations [5]. The p53 mutations frequently develop at six specific residues, known as 'hotspots', located within the DNA-binding domain. These residues are R175, G245, R248, R249, R273, and R282. The majority of missense mutations are located in the DNA-binding domain of p53. Consequently, the mutant protein is unable to activate the majority of its intended genes. This indicates that the wild-type p53 is unable to perform its function. Furthermore, the mutant p53 protein frequently has a dominant negative impact on the remaining wild-type p53. Moreover, mutant p53 develops cancer-causing abilities that impact numerous characteristics, including the transformation from epithelial to mesenchymal cells (EMT), movement, infiltration, spread to other parts of the body, resistance to chemotherapy, growth, cell death, and instability in the genetic material. These functions are completely separate from the wild-type p53. The two primary categories of p53 mutations are as follows: [1] DNA contact defective mutants, which involve alterations in the region of the protein responsible for binding DNA (e.g., R273H, R273C, R248Q, and R248W); and [2] structural defective mutants, which involve changes in the region of the protein necessary for correct folding for example R175H, G245S, R249S, and R282H. Recent research has demonstrated that mutant p53 has the ability to control gene expression and produce cancer-causing effects by acting on certain miRNAs [6]. Despite extensive research efforts, the effect of missense mutations on TP53 structure, function, and interaction networks in the context of HNSCC is not well characterized. Traditional experimental approaches for assessing the functional consequences of TP53 mutations are often labor-intensive, time-consuming, and limited by technical challenges. However, the more recent availability of complex HNSCC associated genomics and genetics datasets allowed computational analyses to revolutionize our ability to predict the effects of genetic variations on protein structure and function in a high-throughput manner [7]. There have been several attempts to categorize mutations based on their impact on causing variations in interactions that could alter protein structure, and these effects could be used to determine their value as prognostic variables. However, these efforts may have limited results due to the diversity of TP53 mutations associated with HNSCC [8]. The Cancer Genome Atlas and other recent studies considering the impact of mutations on the progress and inhibition of HNSCC have provided valuable insights into the molecular pathophysiology of HNSCC development and the response to therapy. Studying the pathogenic mechanism of mutant p53 in HNSCC is crucial for creating individualized treatment approaches for patients [8]. This study aims to assess the impact of missense mutations on TP53-associated HNSCC. Leveraging computational approaches, including protein structure prediction, molecular docking, simulation, and network analysis. This in silico evaluation holds promise for uncovering novel insights into the molecular mechanisms driving HNSCC progression and guiding the development of targeted therapies tailored to the specific genetic landscape of individual tumors.

## 2. Materials and methods

### 2.1. Data sources and data collection

The data for this work was obtained from the TCGA Data Project (https://www.cancer.gov/ccg/research/genome-sequencing/tcga). The TCGA Data Project provides extensive information on genetic changes, gene expression, DNA methylation, and clinical outcomes for various types of cancer. The download was conducted through the Genomic Data Commons Data Portal (https://portal.gdc.cancer.gov/) [9]. Three studies on HNSCC were chosen from cBioPortal (http://www.cbioportal.org). These studies include the TCGA Firehose Legacy, TCGA Nature 2015, and TCGA Pan Cancer Atlas datasets. These investigations incorporated data from 530, 279, and 523 samples, respectively. The genomic profile changes that were examined mutations incloud putative copy number alterations (CNAs), and structural variants. Moreover, the survival

study was analyzed for an altered group, with overlapping samples excluded. In addition, the cBioPortal was utilized to obtain a list of the top 50 genes that exhibit the highest frequency of alterations in HNSCC, including TP53 mutations [10].

## 2.2. Gene expression analysis

The UALCAN tool (https://ualcan.path.uab.edu/) was employed to assess the expression of TP53 and the miRNA-seq (hsa-mir-183, hsa-mir-133b, and hsa-mir-145) in relation to TP53 mutations associated with HNSCC. This analysis was conducted using the TCGA-assembler pipeline [11].

## 2.3. The protein–protein interaction (PPI)

The network among TP53 and the top 50 frequently altered genes was constructed using the STRING (https://string-db.org/) database with a default confidence score threshold of 0.4, which corresponds to medium confidence in the interactions, and visualised visualized using Cytoscape (version 3.8.2) [12].

## 2.4. Missense mutations analysis

**2.4.1. Retrieval of missense mutations.** The TP53 gene (ENSG00000141510) that encodes the tumor protein p53 (Uniprot ID: P04637.4) was obtained from the TCGA database. The missense mutations with significant impact, moderate effect, harmful, and likely damaging effects, as identified in the TCGA database through VEP (https://www.ensembl.org/info/docs/tools/vep/index.html), SIFT (http://siftdna.org/www/SIFT_dbSNP.html), and Polyphen (http://genetics.bwh.harvard.edu/pph2/) were retrieved and analysed by PhD-SNP, PANTHER, and SNPs&GO tools [9].

**2.4.2. Analysis of mutations effect on the TP53.** The functional effects of the missense mutations were analyzed using the PhD-SNP, PANTHER, and SNPs&GO tools [13].

**2.4.3. Oncogenic and phenotypic analysis.** The mutations that were identified as deleterious by a minimum of six software programes were selected for analysis of their oncogenic and phenotypic impacts. The objective of this investigation was to determine whether the somatic point mutations occurring in both the coding and non-coding regions of the cancer genome had a disease-causing effect or were benign. The prediction was generated utilizing CScape and CScape-somatic and FATHMM programs. Subsequently, the oncogenic and driver mutations were selected for comprehensive analysis utilizing diverse software techniques [14].

**2.4.4. Protein stability analysis.** The impact of mutations on protein stability was assessed using the computational tools I-mutant.20, CUPSAT, and MUpro [15].

**2.4.5. Protein evolutionary conservation analysis.** In order to determine the degree of evolutionary conservation of the mutations in the protein sequence, we utilized ConSurf (https://consurf.tau.ac.il) [16].

**2.4.6. Predicting the distribution of mutations on the protein.** The TP53 mutations were mapped using the Mutation 3D tool available at (http://mutation3d.org/) [14].

**2.4.7. Post-translational modification (PTM).** The TP53 protein's tyrosine kinase phosphorylation sites and serine/threonine kinase phosphorylation sites were predicted using the Group-based Prediction System (GPS) 5.0 programe. The GPS-MSP (Methyl-group Specific Predictor) tool was utilized to forecast the methylation site of the protein [17].

**2.4.8. Analysis of pockets with the CASTp tool.** The CASTp tool was utilized to predict the pocket region of the protein and identify any mutations present [17].

**2.4.9. Predicting effects of mutations on the secondary structure of TP53.** The SOPMA server was utilized to predict the impact of mutations on the secondary structure of the TP53 protein, both for the unmutated protein and the mutated protein [18].

**2.4.10. Investigating the structural effects of mutations.** HOPE (https://www3.cmbi.umcn.nl/hope) was used to determine the impact of mutations on protein structure. Furthermore, the UCSF Chimera molecular visualization program

was utilized to compare the wild-type and mutant structures, analyze variations in hydrogen bonding, and detect clashes due to mutations [13].

**2.4.11. Docking analysis.** Docking studies were performed using AutoDock and HDock servers. The AutoDock 3D structure utilized the PDB structure 2AC0 (Chains A & B) as the receptor. The mutated receptor was created by altering residues in the original receptor file using UCSF Chimera 1.16 using the rotamer option. The ligand was provided in the server search box of the HDock Server using the code 2AC0: EF. COOT was used to model the mutant P53 structures with conflicts involving residues G105 and G266E by mutating and resolving different conformations of the residues. Following the creation of the original model, the geometry minimization function in PHENIX was utilized to reduce clashes. The HDOCK/pyDOCK server was utilized to conduct protein-DNA docking with the modelled mutant P53. The receptor files from the previous docking operation were utilized for docking with the pyDOCK server [19].

**2.4.12. Molecular dynamics simulation.** Molecular Dynamics (MD) simulations were conducted on a computational platform featuring a 12th Gen Intel(R) Core(TM) i5-12400F processor, 16.0 GB RAM, a PNY GeForce RTX™ 4060 8GB VERTO™ Dual Fan DLSS 3 GPU, and 64-bit operating system with an x64-based processor architecture. The NAMD software [20] was employed to investigate the dynamic behavior and substrate-bound interactions of proteins. Initial PSF files were generated using Visual Molecular Dynamics (VMD) software [21]. The PSF and PDB files of the protein dimer were solvated using the TIP3P method with a maximum box padding of $5 \times 5 \times 5$ Å$^3$ utilizing VMD 1.9.3 with OpenGL Display. Short-range non-bonded interactions were computed with a 10 Å cut-off distance, while long-range electrostatic interactions were evaluated using the particle mesh Ewald (PME) method [22]. The simulation temperature was maintained at 310 kelvin using the NVE ensemble. Each simulation employed a time step of 2 fs, with a scaling factor and dielectric constant set to 1.0. Visualization and data processing were performed using VMD software. The prepared systems underwent 10,000 steps of energy minimization prior to production runs, which consisted of 100,000,000 steps each with CUDA acceleration enabled using recommended settings; stepsPerCycle, pairlistsPerCycle, margin, outputEnergies, and outputTimings were left undefined. The NAMD configuration specified 4 CPU cores for computational tasks [22].

## 3. Results

### 3.1. Data analysis

The analysis revealed a total of 319 mutations in the TP53 gene. These mutations were categorized into seven types: 78 frameshifts, 146 missense, nine inframe deletions, one intron mutation, six synonymous, 39 stop-gained mutations, and 40 splicing (including donor, acceptor, and region) (Fig 1A and B). In these mutations, 226 substitutions, 24 insertions, and 76 deletions were found S1 Data. Furthermore, 20 genes showed the most frequent distribution of mutations, with the TP53 protein displaying a higher mutation frequency (Fig 2).

### 3.2. Genomic alterations of TP53 in HNSCC patients

Out of 1332 samples, 932 (70%) showed TP53 mutations, which were significantly associated with both the presence of HNSCC (P = 0.0349), indicating that patients with these mutations are more likely to develop the disease, and with reduced progression-free survival (P = 0.0102), suggesting that TP53 mutations may be linked to more aggressive or treatment-resistant forms of HNSCC. This result underscores the importance of TP53 mutations in both the presence and progression of HNSCC, making it a potential target for diagnostics and personalized treatment strategies Table 1.

### 3.3. The TP53 gene mutations and miRNA expression analysis of HNSCC patients

In HNSCC patients, the expression of the mutant TP53 gene was increased, however, this increase had no significant impact on the survival of HNSCC patients Fig 3A and B. This suggests that, while TP53 mutations are common in HNSCC, their overexpression alone does not directly influence survival outcomes. Furthermore, the expression of TP53

**A**

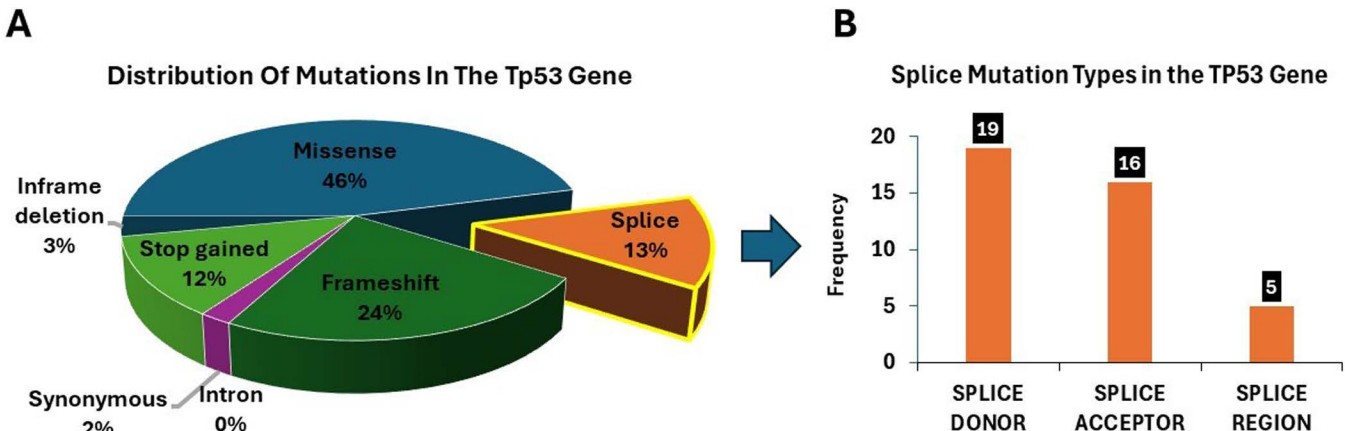

**Fig 1. (A) Pie chart showing the distribution of mutation types identified within the TP53 gene.** The chart highlights the predominance of missense mutations, followed by frameshift and splice-site mutations, with each category represented as a proportion of the total observed mutations. (B) Bar graph presenting the frequency of different splice mutation types in the TP53 gene, categorized into donor site mutations, acceptor site mutations, and mutations within the splice region.

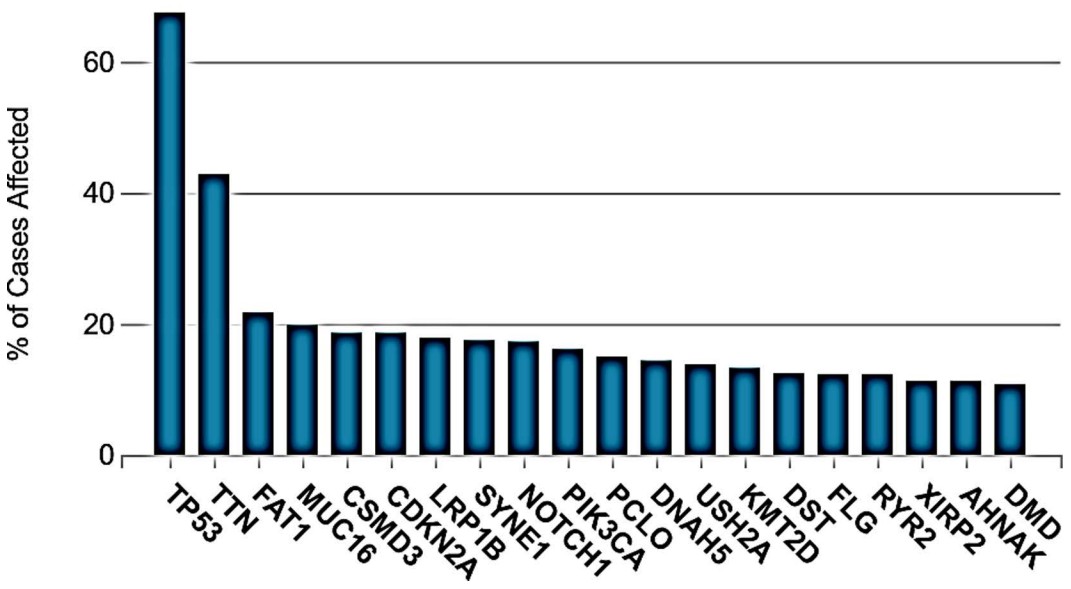

**Fig 2. Graph illustrating the distribution of mutations across 20 genes.** Each bar represents a specific gene, and the height of the bar corresponds to the frequency of mutations identified within that gene. This visualization highlights the variability in mutation density among the analyzed genes, with some genes showing a higher mutation frequency than others.

also increases in Ovarian (OV), Esophageal (ESCA), Rectal Adenocarcinoma (READ), and HNSCC cancers compared to the other 33 tumours studied (Fig 3C), indicating that TP53 dysregulation is a common feature across multiple cancers. Regarding microRNAs (miRNAs), the expression of hsa-mir-183 was elevated in both mutant and non-mutant TP53 groups, but it did not significantly impact survival (P = 0.67), suggesting that its role in HNSCC progression may be limited.

**Table 1. The table shows the significant and non-significant altered group with different parameters.**

| Survival Type | Number of Patients | Altered group | p-Value |
|---|---|---|---|
| Overall | 1207 | 851 | 1.42E-06 |
| Disease Free | 528 | 352 | 5.28E-03 |
| Progression Free | 522 | 357 | 0.0102 |
| Disease-specific | 497 | 338 | 0.0349 |

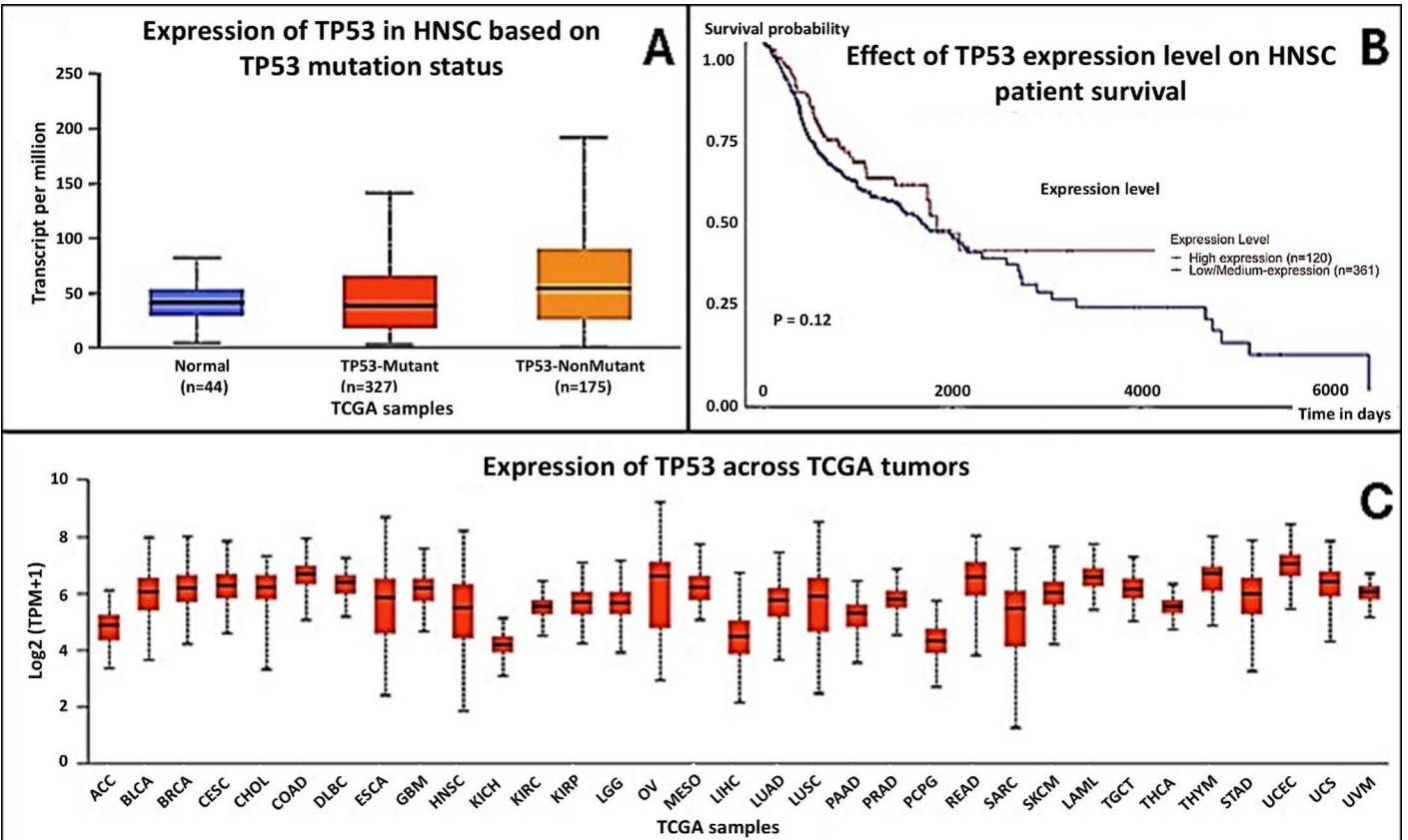

**Fig 3. (A) TP53 expression levels are elevated in TP53-mutant and nonmutant HNSC samples compared to normal tissues.** (B) Kaplan-Meier analysis shows a trend of improved survival in HNSC patients with high TP53 expression, though not statistically significant (P = 0.12). (C) TP53 expression is high across several cancer types, with variation observed between tumor types.

Conversely, the expression of hsa-mir-133b was significantly downregulated in both mutant and non-mutant groups, and this decrease was associated with poorer survival (P = 0.017), identifying hsa-mir-133b as a potential biomarker for monitoring HNSCC progression. Lastly, hsa-mir-145 was upregulated in all patient groups and showed a near-significant correlation with survival (P = 0.056), hinting at its possible role as a prognostic factor S1 Fig.

**Protein-Protein Interaction (PPI):** A PPI network based on the top 50 positively correlated TP53-related genes was created. The result showed that TP53 was related with 23 proteins. Moreover, nine hub genes were identified with TP53, such as CSMD3, FADD, HRAS, FGF4, FGF3, CCND1, MYEDY, CDKN2A, and NOTCH2 S2 Fig. The identification of these nine hub genes in the TP53 interaction network reveals the extensive role TP53 plays in regulating critical cellular

pathways. Genes like HRAS, CCND1, and CDKN2A are well-known drivers of cancer-related processes, including cell cycle progression, and apoptosis. This suggests that mutations in TP53 could disrupt its interactions with these hub genes, potentially leading to unchecked cell growth and survival, which are hallmarks of cancer. Understanding these interactions can provide valuable insights for targeting TP53-related pathways in cancer therapy.

### 3.4. Mutations analysis

**3.4.1. Missense mutations retrieval from TCGA database.** A total of 118 out of 145 mutations were found to have significant impact, moderate effect, harmful, and likely damaging effects by VEP, SIFT, and Polyphen were retrieved S2 Data and analysed by PhD-SNP, PANTHER and, SNPs&GO.

**3.4.2. Mutations analysis using PhD-SNP, PANTHER and, SNPs&GO.** The analysis of 118 mutations using PhD-SNP, PANTHER and, SNPs&GO showed that 90 mutations were found disease by the three software S3 Data.

**3.4.3. Oncogenic analysis of mutations.** The CScape analysis revealed that 13 out of 90 missense mutations possess oncogenic properties S4 Data, which six mutations (R273C, G105C, G266E, Q136H/P, R280G) were identified as driver mutations and nine as passenger mutations, using the CScape-somatic method Table 2.The driver mutations highlight their critical role in tumorigenesis, as driver mutations are known to confer a selective growth advantage to cancer cells.

**3.4.4. Phenotypic analysis of mutations using coding variant, cancer.** The phenotypic effects of the mutations indicated that R273C, G105C, G266E, Q136P, Q136H, and R280G were categorized as cancer-causing mutations Table 3. By classifying R273C, G105C, G266E, Q136P, Q136H, and R280G as phenotypically significant, this result highlights their contributions to the oncogenic processes. These mutations may disrupt normal protein function, leading to uncontrolled cell proliferation, evasion of apoptosis, and other hallmarks of cancer.

**3.4.5. Measuring the structural stability of proteins.** The prediction of the impact of TP53 mutations on stability revealed that the six mutations decrease protein stability as determined by I-MUTANT 2.0, MUpro, and CUPSAT triple software Table 4. Reduced protein stability could lead to misfolding or degradation of the TP53 protein, impairing its role in regulating cell cycle and apoptosis. This destabilization may facilitate oncogenic processes by allowing cells to evade normal growth control mechanisms, ultimately contributing to tumor progression.

**Table 2. Oncogenic nature of mutations predicted using CScape and CScape-somatic.**

| Chr, Position, DNA change | AA change in TP53 | Predication by CScape | Predication by CScape-somatic | Message | Coding Score |
|---|---|---|---|---|---|
| 17, 7673803 G, A | R273C | Oncogenic | 0.518551 | Driver | 0.649700 |
| 17,7673823 C, T | G266E | Oncogenic | 0.756660 | Driver | 0.502703 |
| 17, 7674885 C, T | V216M | Oncogenic | 0.650734 | Passenger | 0.336012 |
| 17, 7673782 T, C | R280G | Oncogenic | 0.587732 | Driver | 0.598344 |
| 17, 7673781C, T | R280K | Oncogenic | 0.724274 | Passenger | 0.448332 |
| 17,7676055 C, A | G105C | Oncogenic | 0.512638 | Driver | 0.776365 |
| 17, 7675205 T, G | Q136P | Oncogenic | 0.695476 | Driver | 0.771776 |
| 17, 7673781 C, T | R280K | Oncogenic | 0.724274 | Passenger | 0.448332 |
| 17,7675204 T, A | Q136H | Oncogenic | 0.695476 | Driver | 0.771776 |
| 17,7674947 A, T | I195N | Oncogenic | 0.795476 | Passenger | 0.266789 |
| 17, 7674211, A,T | I251N | Oncogenic | 0.904874 | Passenger | 0.193159 |
| 17,7674917, T, C | Y205C | Oncogenic | 0.646447 | Passenger | 0.322967 |
| 17,7674947, A, G | I195T | Oncogenic | 0.619239 | Passenger | 0.348914 |
| 17, 7673824, C, T | I251N | Oncogenic | 0.693675 | Passenger | 0.420640 |
| 17,7669614, C. G | D393H | Oncogenic | 0.746424 | Passenger | 0.425802 |

**Table 3. Result of the phenotypic analysis.**

| Mutations | Classification | Score |
|---|---|---|
| R273C | CANCER | −9.58 |
| G105C | CANCER | −10.04 |
| G266E | CANCER | −9.22 |
| Q136P | CANCER | −9.58 |
| Q136H | CANCER | −9.38 |
| R280G | CANCER | −8.20 |

**Table 4. Effect of mutations on protein stability predicted by I-MUTANT 2.0, MUpro and CUPSAT.**

| Mutations | Effet | DDG | MUpro | Score | CUPSAT | Score |
|---|---|---|---|---|---|---|
| R273C | Decrease | −1.94 | Decrease | −0.064804414 | Destabilizing | −1.41 |
| G105C | Decrease | −0.61 | Decrease | −0.38230284 | Destabilizing | −1.45 |
| G266E | Decrease | −0.07 | Decrease | −0.71022011 | Destabilizing | −1.1 |
| Q136P | Decrease | −0.61 | Decrease | −0.95908148 | Destabilizing | −1.04 |
| Q136H | Decrease | −0.52 | Decrease | −0.8118986 | Destabilizing | −0.55 |
| R280G | Decrease | −2.53 | Decrease | −1.4659608 | Destabilizing | −1.26 |

**3.4.6. Analysis of evolutionary conservation.** The outcome generated by the ConSurf web server indicated that the mutations R273C, Q136H, Q136P, G105C, G266E, and R280G are functional residues that are highly conserved and exposed S3 Fig. The result suggest that these mutations are critical to the structure and function of the TP53 protein. Their location in highly conserved regions suggests that alterations could disrupt essential biological processes. Additionally, the exposure of these residues indicates potential interactions with other proteins or DNA, highlighting their significance in TP53's tumor-suppressing activities.

**3.4.7. Distribution of missense mutations on the TP53 protein.** The distribution of mutations R273C, G105C, G266E, Q136P, Q136H, and R280G in the P53 domain (amino acids 95–289) of the TP53 protein was seen in the result Fig 4. The presence of these mutations in the P53 domain underscores their potential to disrupt the protein's DNA-binding activity, impairing its ability to regulate genes crucial for cell cycle control and apoptosis.

**3.4.8. Post-translational modification analysis and methylation site analysis.** Analysis of post-translational modification sites showed that none of the mutations occur at phosphorylation sites. However, the G105C mutation is in close proximity to S106, and the G266E mutation is near S269 S4 Fig, the absence of mutations at phosphorylation sites, suggests that these changes may not directly impact TP53's phosphorylation-dependent regulation. However, the proximity of G105C to S106 and G266E to S269 could affect the phosphorylation status of nearby residues, potentially altering regulatory mechanisms. Furthermore, the funding indicated that R280G and R273 were located at the methylation site, the presence of R280G and R273C at methylation sites indicates that these mutations may disrupt methylation processes, which are essential for regulating protein function and interactions.

**3.4.9. Pocket site prediction using the CASTp 3.0 server.** The analysis revealed that R273C, Q136H/P, and R280G were situated within the protein's pocket region S5 Fig. The mutations located in the pocket region of the TP53 protein are significant because this area is essential for DNA and protein interactions. Alterations in this region can reduce binding affinity and specificity, impairing TP53's ability to regulate genes involved in cell cycle control and apoptosis. This disruption may contribute to oncogenic processes linked to TP53 mutations, highlighting the need for further investigation and potential therapeutic targeting of these alterations.

 

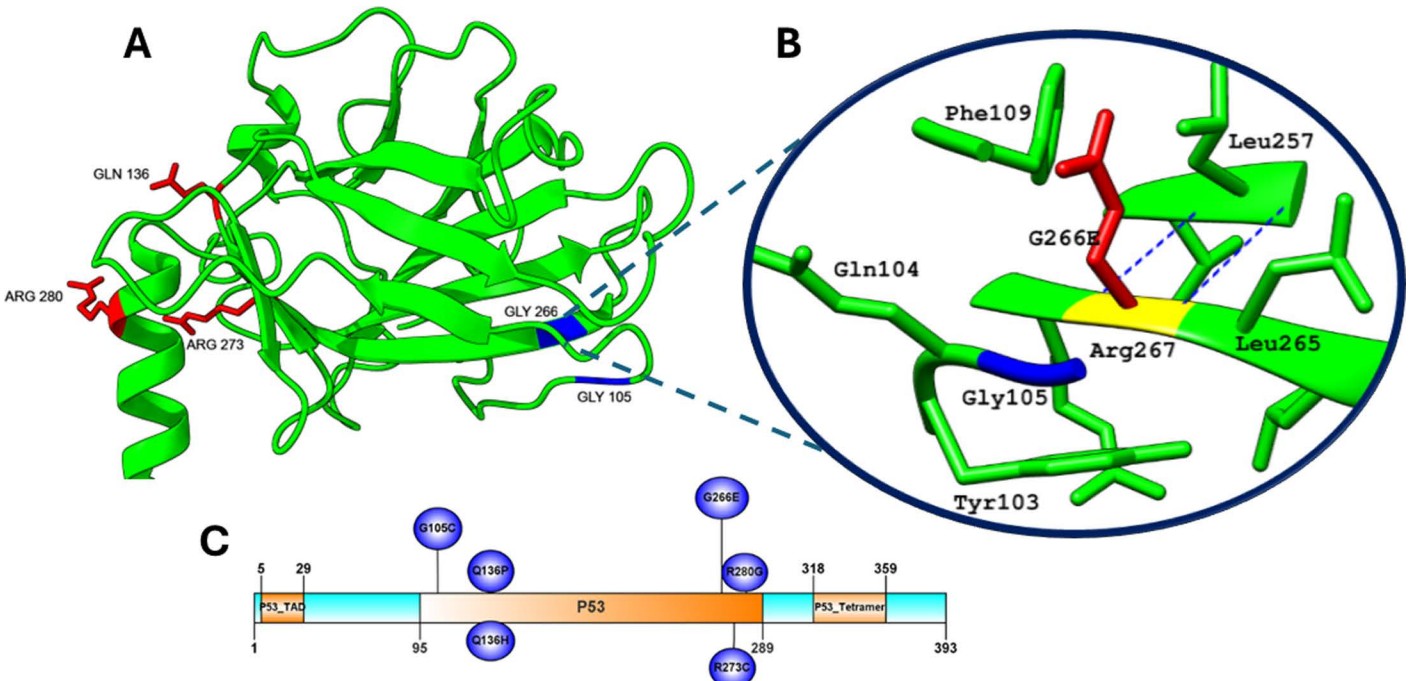

**Fig 4. (A) The distribution of mutations within the 3D structure of the TP53 protein.** (B) Structural alteration of the wild type residue G266E by Project Hope. The wild type residue is presented as yellow and the mutant residue is shown in red. (C) The illustration represented mutations occurring in the P53 domain of the TP53 protein.

**3.4.10. Secondary structure predication.** The study demonstrated that the six mutations altered the secondary structure, including the alpha helix, extended strand, turn, and coil. The alterations can affect the stability and functionality of the TP53 protein. Changes in elements like the alpha helix and extended strands may impair the protein's ability to bind DNA and regulate gene expression. Among these mutations, G105C and R280G had a greater impact on these structural changes Table 5.

**3.4.11. Structural effect of missenes mutations on TP53 protein.** The Project HOPE server identified that the R273C, G105C, G266E, Q136P, Q136H, and R280G mutations showed differences in size, hydrophobicity, and charge compared to the wild-type. Moreover, these mutations affect the hydrogen bonds and salt bridges formed by the wild-type. These mutations were identified within the domain and conserved region, and they impact the structure and function of the protein S1 Table.

**Table 5. The result of the secondary structure prediction of native and six mutant proteins of TP35 protein using the SOPMA server.**

| Mutation | Alpha helix | Extended strand | Beta turn | Random coil |
|---|---|---|---|---|
| Wild type | 78 (19.85%) | 71 (18.07%) | 14 (3.56%) | 230 (58.52%) |
| R273C | 75 (19.08%) | 71 (18.07%) | 11 (2.80%) | 236 (60.05%) |
| G105C | 76 (19.34%) | 66 (16.79%) | 9 (2.29%) | 242 (61.58%) |
| G266E | 74 (18.83%) | 70 (17.81%) | 10 (2.54%) | 239 (60.81%) |
| Q136P | 78 (19.85%) | 68 (17.30%) | 11 (2.80%) | 236 (60.05%) |
| Q136H | 77 (19.59%) | 67 (17.05%) | 11(2.80%) | 238(60.56%) |
| R280G | 75 (19.08%) | 67(17.05%) | 8 (2.04%) | 243 (61.83%) |

**3.4.12. H-bonds analysis of wild and mutant types.** The UCSF Chimera molecular visualization and analysis software was utilized to perform H-bond and clash analysis on both wild-type and mutant type "Figs 5,6". The analysis revealed a difference in the number of hydrogen bonds in the mutant structure across all mutations. For the G266E mutation, there are 20 hydrogen bonds in the environment of the wild type, compared to 25 hydrogen bonds in the mutant type Fig 5A and B. Similarly, for the G105C mutation, the wild type environment contains 6 hydrogen bonds, increasing to 8 hydrogen bonds in the mutant Fig 5C and D. For the R273C mutation, the wild type has 15 hydrogen bonds, while the mutant type shows 9 hydrogen bonds Fig 5E and F. In the R280G mutation, the wild type environment contains 12 hydrogen bonds, whereas the mutant type exhibits 11 hydrogen bonds Fig 5G and H. Lastly, for the Q136P mutation, the wild type has 9 hydrogen bonds, with the Q136P mutant showing 5 hydrogen bonds. Another variant, Q136H, maintains 9 hydrogen bonds in its environment Fig 5I and J. These alterations in hydrogen bonding and structural integrity could significantly impair the tumor-suppressing functions of TP53, emphasizing the need for targeted strategies to mitigate these effects. Additionally, the G105C, G266E and Q136H mutations showed a prevalence of clashes Fig 6. The increased prevalence of clashes associated with G266E, G105C, and Q136H indicates potential steric hindrance, which could further compromise the protein's ability to bind to its targets effectively.

**3.4.13. Docking result.** The docking outcomes generated by both servers, HDOCK and pyDOCK, indicated that mutations, including those affecting residues not directly interacting with DNA, decreased binding score values Table 6. Moreover, Both HDOCK (turquoise) and pyDOCK (yellow) web servers achieved excellent redocking results compared to the original structure (magenta), with rmsd values of 0.3273 and 1.1218, respectively Fig 7. The reduction in binding score values due to mutations—regardless of their direct interaction with DNA—indicates a negative impact on the binding affinity of the TP53 protein, potentially impairing its tumor-suppressing functions. While the strong redocking results affirm the reliability of the docking predictions, variations in rmsd values suggest differing influences of mutations on docking conformations.

**3.4.14. Molecular dynamics simulation.** During the 200 ns simulation, Apo-TP53 exhibited the highest average RMSD value of 26.20 Å among the simulated complexes, suggesting that in the absence of the DNA substrate, the TP53 protein cannot maintain its dimeric form. The average RMSD values for the TP53 mutants were determined as 3.74 Å for the G105C mutant, followed by 4.79 Å for Q136H, and 4.86 Å for R280G Fig 8A. All tested mutations resulted in a relative increase in average RMSD compared to wild-type TP53 (3.48 Å), suggesting potential destabilization of the protein-DNA complex structures. Root-mean-square fluctuation (RMSF) measures the variability in the position of individual residues over time, providing insights into the flexibility and dynamic behavior of regions within the protein structure. RMSF analysis showed that each mutation affects average RMSF values differently, with G105C and Q136H causing a slight decrease (indicating increased rigidity). At the same time, R280G leads to a significant increase in RMSF (indicating greater flexibility) Fig 8B. Hydrogen bond analysis further quantified the frequency and types of hydrogen bonds formed between various TP53 forms and DNA substrates, with R280G exhibiting the lowest hydrogen bond formation rate at 0.024 ns$^{-1}$, followed by the Q136H mutant (0.031 ns$^{-1}$), G105C (0.035 ns$^{-1}$), and the wild-type at 0.0360 ns$^{-1}$ Fig 8C. The solvent-accessible surface area (SASA) analysis illustratedthe relationship between SASA and time for TP53 and its mutants Fig 8D. Lastly, the radius of gyration (Rg) analysis,which evaluates the compactness of the protein-substrate complex, showed that at the end of the simulation, Apo-TP53 exhibited the highest average Rg value (45.07 Å), Q136H (30.11 Å), G105C (28.91 Å), wild-type (28.43 Å), and R280G (27.44 Å) Fig 8E. Table 7.

## 4. Discussion

Various efforts have been made to classify TP53 mutations according to their influence on inducing changes in interactions that may modify protein structure. These effects can be utilized to assess their significance as indicators of disease progression. However, these efforts may have been limited due to the diversity of TP53 mutations. The Cancer Genome Atlas and other recent studies examining the impact of mutations on HNSCC progress and inhibition have provided

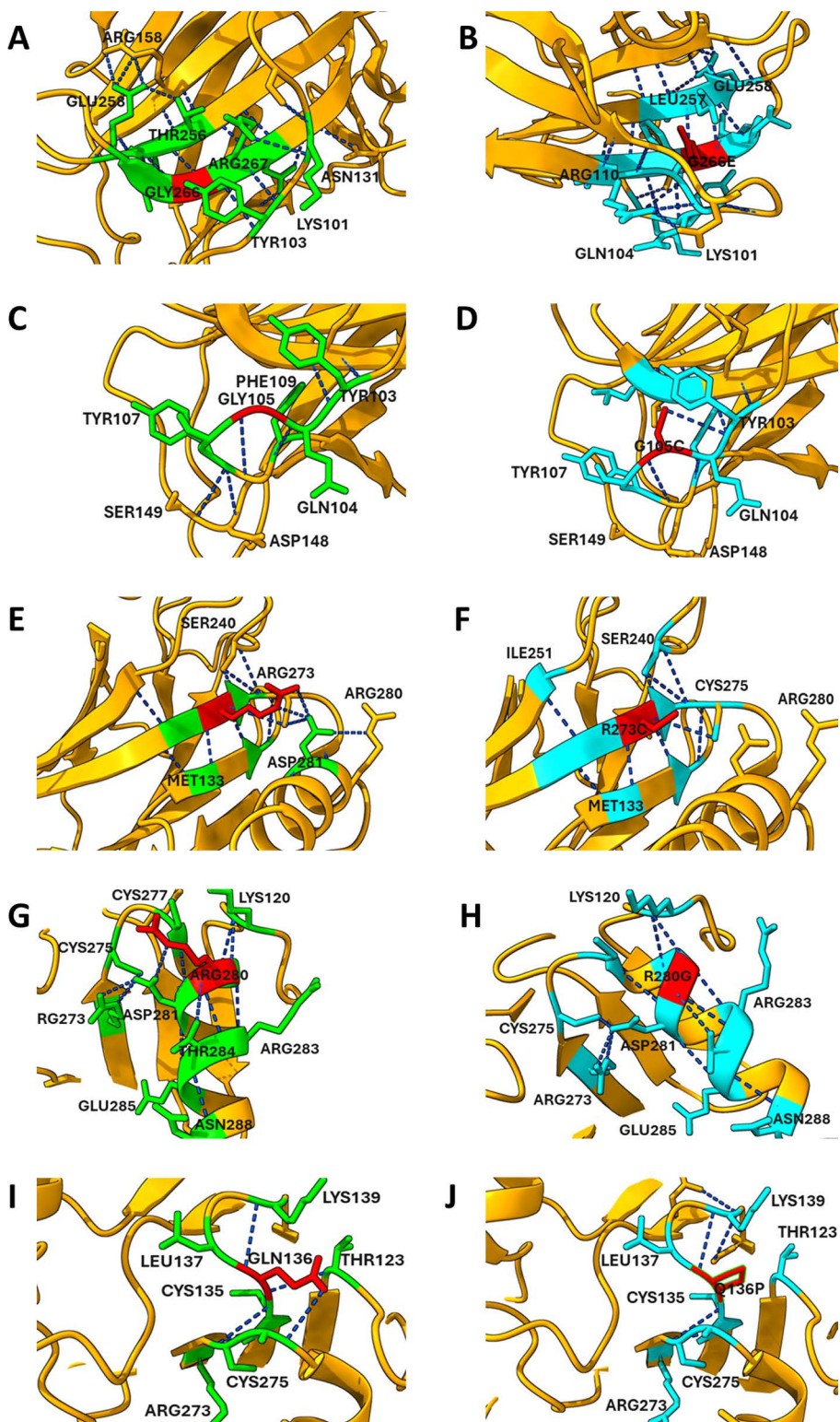

**Fig 5. Structural alterations in hydrogen bond networks of the TP53 protein caused by mutations G266E (A-B), G105C (C-D), R273C (E-F), R280G (G-H), and Q136P (I-J).** Wild-type structures (left panels) and mutant structures (right panels) highlight the disruption of key hydrogen bonds (blue dashed lines) and changes in residue interactions (green sticks). Mutated residues are marked in red, demonstrating destabilizing effects on the TP53 conformation and DNA-binding ability. Yellow ribbons represent the protein backbone, emphasizing structural distortions caused by each mutation.

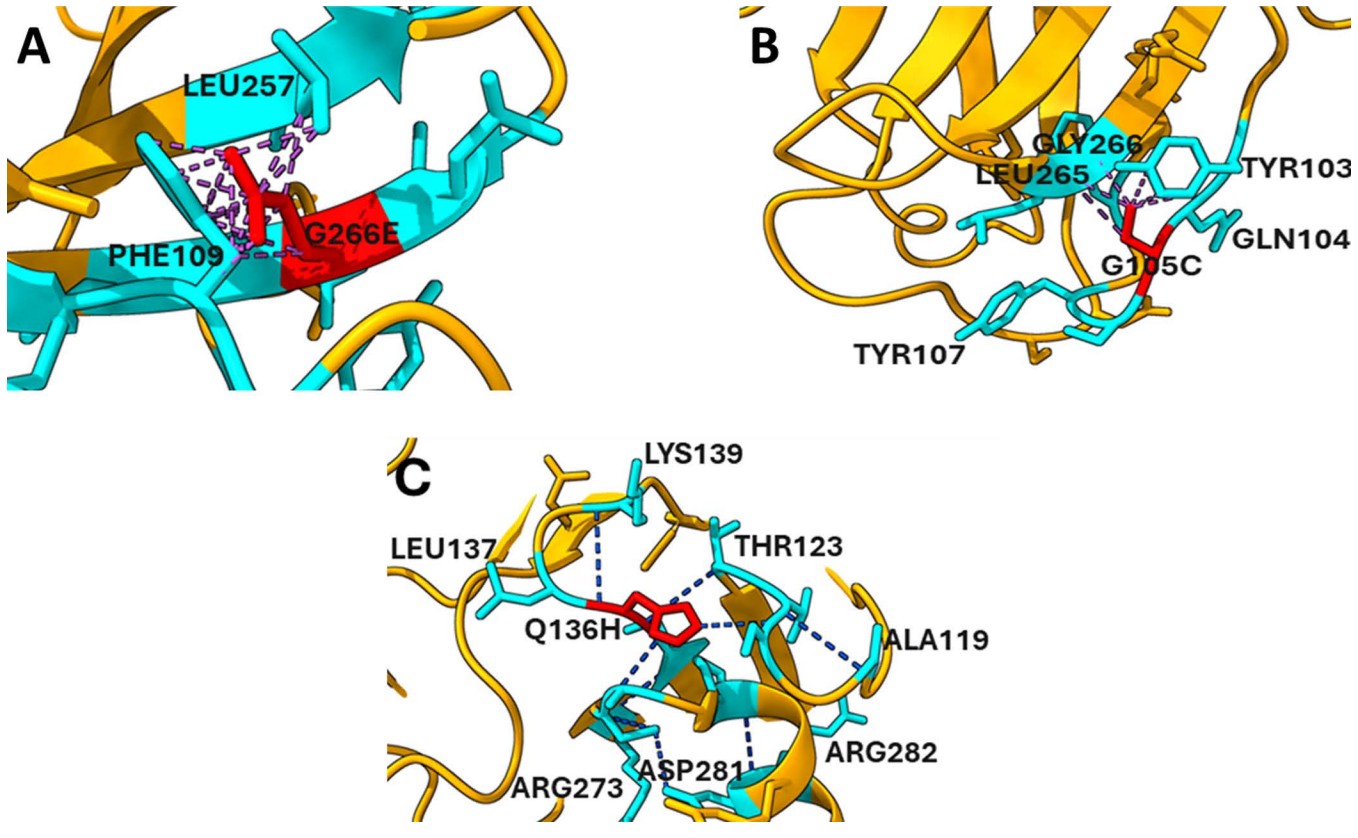

**Fig 6. Structural clashes in TP53 mutants: (A) G266E mutation causes steric clashes with PHE109 and LEU257, destabilizing the structure.** (B) G105C mutation disrupts interactions with residues LEU265, TYR103, TYR107, and GLN104. (C) Q136H mutation alters interactions with nearby residues (ALA119, THR123, LEU137, LYS139, ASP281, ARG273, ARG282), affecting protein stability.

valuable insights into the molecular pathophysiology of HNSCC development and the response to therapy [23]. In this study, we used bioinformatics tools to investigate the impact of mutations on the expression of the TP53 gene and miRNA on the survival of patients with HNSCC, and to analyze missense mutations found in the TP53 protein from the TCGA database, in order to anticipate on the protein's function, structure, and stability. In this study, the findings showed that missense mutations are the most frequently observed type among six different mutation types, including frameshift mutations and inframe deletions in HNSCC, based on data from the Cancer Genome Atlas (TCGA). Of the 20 genes, the TP53 gene showed the highest frequency of mutations. As a result, these mutations affect both the protein's structure and function, and the TP53 mutations have a significant impact on HNSCC development.

The analysis results for 1332 cases using cBioPortal revealed that 70% of patients with HNSCC have genomic mutations in the TP53 gene. The group with this alteration has a significant correlation with both the disease's presence (P = 0.0349) and its progression (P = 0.0102). The significant correlation with progression-free status may indicate that these alterations could be biomarkers for predicting disease progression or response to treatment. Therefore, understanding the alterations associated with the disease can lead to the development of targeted therapies aimed at these specific changes. The results revealed that mutant TP53 increased the expression levels of TP53, but showed no significant association with patient survival. Therefore, these mutations can disrupt the normal function of TP53, leading to uncontrolled cell proliferation and tumor development. Moreover, the findings revealed a decrease in the expression of hsa-mir-133b in both mutant and non-mutant TP53 proteins. This decrease was found to have a significant association with patient

**Table 6.** P53 mutant docking results (HDOCK and pyDOCK servers).

| Mutation | HDOCK score | pyDOCK score |
|---|---|---|
| No mutation | −392.01 | −170.74 |
| R273C | −352.39 | −141.805 |
| G105C | −355.37 | −164.618 |
| G266E | −355.47 | −158.895 |
| R280G | −326.63 | −147.482 |
| Q136P | −387.24 | −166.966 |
| Q136H | −387.38 | −165.582 |

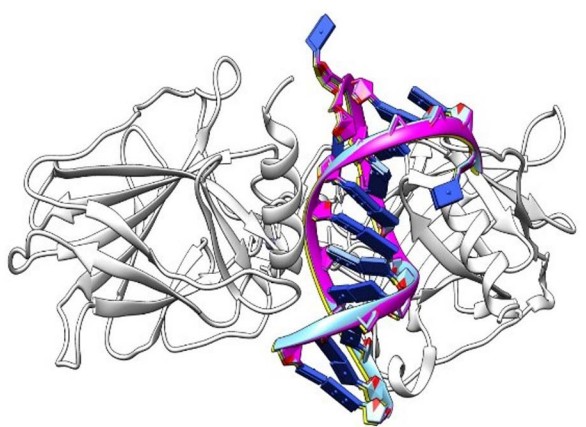

**Fig 7. Redocking results showing the interaction between the TP53 protein (white ribbon) and its DNA dodecamer ligand (magenta and blue) from PDB ID: 2AC0, highlighting the binding conformation.**

survival. The reduced expression of miR-133b in TP53-related cancers has a significant impact on survival due to its role as a tumor-suppressor miRNA. In cancer, miR-133b is downregulated, which affects processes critical to cancer progression. This specific miRNA normally targets genes involved in cell proliferation, migration, and invasion, such as Sox9, c-MET, and WAVE2. When miR-133b expression is low, these targets are less regulated, allowing cancer cells to grow, invade, and metastasize more freely. Furthermore, miR-133b's downregulation is associated with higher tumor grade and poor prognosis in breast cancer, suggesting its levels could serve as a potential prognostic marker [24]. In a previous study, it was shown that hsa-mir-133b is a promising biomarker for oral squamous cell carcinoma [25].

The PPI based on the top 50 positively correlated TP53-related proteins, revealed nine hub proteins closely associated with TP53: CSMD3, FADD, HRAS, FGF4, FGF3, CCND1, MYEDY, CDKN2A, and NOTCH2. These hub proteins are implicated in various cellular processes and pathways, including cell proliferation, apoptosis, cell cycle regulation, and signalling pathways [26].

After analyzing the mutations in TP53 and their effects on expression, and protein-protein interactions with HNSCC, it is crucial to identify the most harmful mutations and their impact on the structure and functions of the TP53 protein linked to HNSCC. Analysis of 146 missense mutations in the TCGA database revealed that six mutations, R273C, G105C, G266E, Q136P, Q136H, and, R280G significantly impact the structure and function of the TP53 protein, leading to phenotypic effects associated with cancer. These driver mutations play a crucial role in the development and progression of cancer by promoting malignant transformation in cells [27]. Their identification is vital for enhancing our understanding of cancer

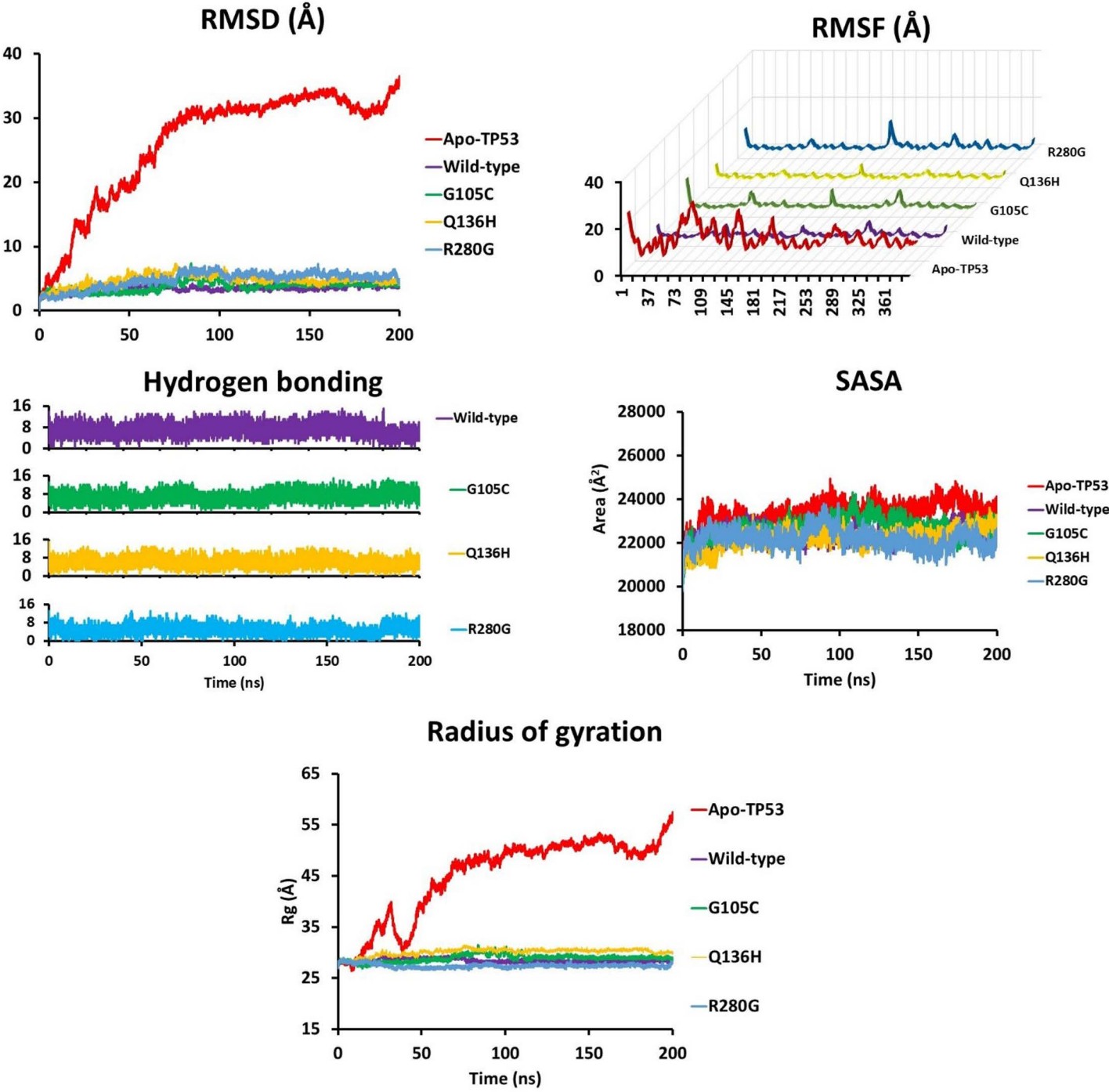

**Fig 8. Molecular dynamics simulation results for Apo-TP53 (red), wild-typeTP53.** DNAcomplex(purple), and the complexes between DNA substrate and TP53 mutants; G105C(green), Q136H (yellow), R280G (blue).(A) Root Mean Square Deviation (RMSD) analysis, (B) RootMeanSquare Fluctuation(RMSF) analysis, (C) Hydrogen bonding analysis, (D) Solvent accessible surface area analysis, and (E) Radius of gyration on analysis.

**Table 7. The result of molecular dynamics simulation.**

| Parameter | Structure | 200 ns |
|---|---|---|
| RMSD | Apo-TP53 | 26.20 |
| | Wild-type TP53 | 3.48 |
| | G105C | 3.74 |
| | Q136H | 4.79 |
| | R280G | 4.86 |
| RMSF | Apo-TP53 | 13.09 |
| | Wild-type TP53 | 2.17 |
| | G105C | 2.15 |
| | Q136H | 2.07 |
| | R280G | 2.37 |
| HBond | Wild-type TP53 | 7.10 |
| | G105C | 6.95 |
| | Q136H | 6.23 |
| | R280G | 4.86 |
| SASA | Apo-TP53 | 23410.07 |
| | Wild-type TP53 | 22370.87 |
| | G105C | 22654.58 |
| | Q136H | 22227.30 |
| | R280G | 22202.49 |
| Rg | Apo-TP53 | 45.07 |
| | Wild-type TP53 | 28.43 |
| | G105C | 28.91 |
| | Q136H | 30.11 |
| | R280G | 27.44 |

biology and informing targeted therapeutic strategies, thus facilitating precision medicine to improve patient outcomes. The ConSurf tool's analysis revealed that all mutations are functionally relevant, highly conserved, and exposed mutations located within the TP53 protein's P53 domain (amino acids 95–289). These mutations can disrupt the protein's structure, function, or its interactions with other molecules, leading to a reduced in its tumor suppressor activity. This impairment is critical as it contributes to the initiation and advancement of cancer, positioning these mutations as specific targets for therapeutic interventions aimed at restoring TP53 function or mitigating the effects of its loss in tumorigenesis.

The proximity of the G105C and G266E mutations to key phosphorylation sites (S106 and S269, respectively) suggests that, while they may not directly alter these phosphorylation sites, they could still impact the regulatory environment of nearby residues. This influence could affect the phosphorylation status of these sites or the interactions TP53 has with regulatory proteins, ultimately impacting its function. Additionally, the presence of the R280G and R273 mutations at methylation sites highlights the potential for disrupted regulatory mechanisms. Methylation, particularly of arginine residues, plays a role in controlling TP53's stability, interactions, and activity as a tumor suppressor. Mutations at these methylation sites could impair these regulatory functions, thereby affecting TP53's capacity to suppress tumor development [28]. This analysis underscores the importance of understanding PTMs in TP53's interactions and activity regulation, as changes here may contribute to the impaired tumor-suppressive functions seen in cancer-associated mutations.

The pocket region of TP53 is critical for its interactions with other proteins and DNA, regulating processes such as cell cycle control, DNA repair, and apoptosis. Mutations within this region can disrupt these interactions, leading to dysregulation of TP53-mediated pathways and contributing to cancer development and progression [29]. In this study, R273C, Q136P/H, and

R280G mutations found within the protein's pocket region which suggest these mutations have impact on cancer development and progression. The study's findings that the R273C, Q136P/H, G105C, G266E, and R280G altered the secondary structure of TP53, including the alpha helix, extended strand, turn, and coil, suggest significant structural perturbations in the TP53 protein. These alterations can impact TP53's stability, folding, and functional interactions with other proteins and DNA [29]. Of particular note, the mutations G105C and R280G were identified as having a greater impact on these structural changes compared to the other mutations. This suggests that these mutations may induce more pronounced alterations in the overall conformation of the TP53 protein, potentially leading to functional consequences that contribute to cancer development and progression. The comparison of H-bonds between mutant and wild-type TP53 proteins reveals differences in their interactions and stability. H-bonds play a crucial role in maintaining protein structure and stability. They also mediate interactions with DNA and other molecules [29]. The observed differences in the number of H-bonds between mutant and wild-type TP53 proteins suggests that the mutations may disrupt or alter the H-bonding network within the protein structure. These alterations could affect the stability and conformational dynamics of TP53, potentially impacting its function as a tumor suppressor. Furthermore, the identification of clashes between residues around the G105C and G266E mutations indicates steric hindrance or clashes between neighboring atoms in the protein structure. These clashes may result from bulky or incompatible residues introduced by mutations, resulting in structural distortions and potential functional consequences. The HOPE result revealed that the mutations disrupt the structural integrity of the protein by altering size, charge, hydrophobicity, and key interactions like hydrogen bonds and salt bridges. These changes can lead to protein misfolding, instability, and degradation. Such mutations, particularly in conserved and functional regions, can lead to diseases or disorders if the protein's normal function is compromised.

When compared to the non-mutated P53, the docking results showed that the presence of a mutation led to a noticeable drop in the binding affinity. Suggesting that the interaction between the P53 binding site and its DNA substrate was probably weaker. The R273C, R280G, C105G, Q136H/P, and G266E mutations affect DNA binding to P53, as shown by both the HDOCK and pyDOCK analyses. This observation is significant as residues R273C, R280G, C105G, and G266E are identified as one of the key residues directly involved in contacting DNA through an intricate network of hydrogen bonds and play a crucial role in the docking of P53 to DNA.

To evaluate the impact of TP53 mutations on substrate binding, we analyzed the root mean square deviation (RMSD), root mean square fluctuation (RMSF), number of hydrogen bonds, radius of gyration (Rg), and solvent accessible surface area (SASA). RMSD calculations were performed to determine the degree of deviation of substrate-bound TP53 frames against the top reference frame from the minimization step throughout the simulation. Throughout the simulation, RMSD trajectory analysis revealed significant backbone shifts in all TP53 systems. Substrate binding reduced RMSD values, indicating notable conformational stabilization of the TP53 dimer. Without DNA, the TP53 dimer could not maintain its functional assembly. However, mutations reduced the backbone flexibility of the protein dimer upon DNA substrate binding compared to the wild-type, suggesting that these mutations contribute to destabilization of the TP53 dimer-DNA complex, with R280G being the most disruptive. The averaged RMSF, which assesses the deviation of individual residues, showed an decrease in overall RMSF values upon DNA substrate binding to TP53 relative to its apo counterpart. However, except for R280G, point mutations decreased overall RMSF values, hinting at a slight decrease in flexibility. Hydrogen bond analysis demonstrated a decreased rate of hydrogen bond formation in TP53 mutants, indicating a looser interaction between the mutated TP53 and the DNA substrate. SASA analysis, which predicts the extent of conformational changes upon TP53-DNA interaction, indicated that DNA binding to wild-type TP53 decreased the protein SASA value by 4.43%. Conversely, mutation in the TP53 protein decreases the SASA value even further with R280G showing the highest reduction (-5.16%), followed by Q136H (5.05%). On the other hand, although G105C (3.23%) showed a decrease in SASA compared to the apo structure, it resulted in a slight increase in SASA relative to the wild-type (3.23% vs 4.43%). Reduction in the values suggests that TP53 mutations result in a more condensed TP53-substrate assembly, minimizing the impact of surrounding water molecules. The radius of gyration (Rg) analysis, which measures the compactness of a protein structure, showed that TP53 mutations (except R280G) decrease the compactness of the protein-substrate complex compared

to the wild-type TP53, aligning with the RMSD, RMSF, and SASA results. Overall, molecular dynamics analysis indicates that TP53 mutations confer a gain of function, activating mutant p53 expression and enhancing the survivability of cancerous cells. This gain of function potentially enables the mutant cells to efficiently perform various growth and survival-related functions such as migration, invasion, and metastasis [30].

In vitro studies lend support to our computational analysis, providing experimental evidence for the predicted effects of TP53 mutations. For instance, Joerger and Fersht (2016) demonstrated that certain TP53 mutations destabilize the protein's core structure, impairing DNA binding and diminishing tumor-suppressor functions findings consistent with our computational predictions regarding structural disruptions and functional loss [31].

The G105C and Q136H/P mutations in TP53 are distinct from other well-characterized mutations such as R175H, R273H, Y220C, R196*, and R213*. While the latter mutations are either "hotspot" mutations or result in significant structural alterations, the G105C and Q136H/P mutations occupy less commonly mutated regions, potentially leading to unique impacts on p53's structure and function. For example, R175H and R273H mutations are known for their association with aggressive cancer phenotypes and are frequently located in the DNA binding domain, disrupting p53's interaction with DNA and its tumor-suppressing function. Similarly, Y220C causes structural changes that can destabilize p53, though it is also targetable by certain small molecules designed to restore its function. In contrast, the truncating mutations R196* and R213* result in incomplete p53 proteins that lack critical domains, leading to a complete loss of tumor-suppressive function and promoting cancer progression. G105C and Q136H/P differ in that they may affect p53 stability or protein-protein interactions in ways that do not align with these established mechanisms of dysfunction. This positional uniqueness could reveal alternative pathways of p53 inactivation or novel gain-of-function effects, suggesting that G105C and Q136H/P represent underexplored opportunities for therapeutic intervention distinct from traditional TP53-targeted strategies [32]. The study calls for experimental validation of the findings, development of therapies targeting key TP53 mutations, integration of mutation profiling into clinical practice for personalized treatment, and expanded analysis across diverse populations for broader applicability

## 5. Conclusion

This study indicated that the mutations in the TP53 protein linked to HNSCC have a significant impact on the development and progression of the disease. The analysis of the samples shows that alterations in 70% of the samples are significantly associated with both the presence of the disease and progression-free status, suggesting a meaningful relationship between these alterations and disease outcomes.The in silico analysis of TP53 mutations in HNSCC reveals significant alterations in TP53 and miRNA expression, and hsa-miR-133b shows promise as a novel biomarker for monitoring HNSCC. Moreover, the mutations R273C, G105C, G266E, Q136H/P, and R280G in TP53 significantly impact its structure, function, and interactions with DNA and other molecules. These alterations contribute to oncogenic processes by disrupting normal TP53-mediated cellular functions, thereby promoting cancer development and progression.The identification of R273C, G105C, G266E, Q136H/P, and R280G as driver mutations underscores their pivotal role in cancer pathogenesis. Therefore, understanding these mutations enhances the development of targeted therapies and personalized medicine approaches. The finding of the R273C and R280G mutations, which are positioned in the methylation region, will provide new insights into the molecular mechanism behind these alterations. Furthermore, the impact of the Q136H, G105C, and R280G mutations on average RMSD, RMSF, hydrogen bond count, and Rg values relative to wild-type TP53 varies depending on the specific mutation and simulation conditions, indicating a complex interplay between the mutations and the DNA substrate. These findings provide insights into how these structural metrics correlate with stability, flexibility, and overall conformational changes in TP53 and its mutants under simulation conditions. The Q136H, G105C, and R280G mutations identified through molecular dynamics analysis have significant potential as novel diagnostic biomarkers. Their presence, stability, and impact on TP53 function can provide valuable insights into disease progression and treatment response. Integrating TP53 mutation analysis into clinical practice can enhance cancer diagnosis, facilitate personalized treatment strategies, and ultimately improve patient outcomes by providing valuable insights into disease progression and treatment response

## Study limitations

1. **Experimental Validation:** The study is based entirely on in silico analyses, and no experimental validation was conducted to confirm the computational predictions.

2. **Sample Diversity**: The dataset primarily utilizes publicly available TCGA data, which may not represent all demographic and geographic populations affected by HNSCC.

## Supporting information

**S1 Data.  Type of Mutations.**
(XLSX)

**S2 Data.  Extraction of demaging missense mutations from the TCGA Database.**
(XLSX)

**S3 Data.  Computational analysis of missense mutations using PhD-SNP, PANTHER and, SNPs&GO.**
(XLSX)

**S4 Data.  Oncogenic Classification of TP53 missense mutations.**
(XLSX)

**S1 Table.  The result of structural and function change in Tp53 using HOPE server.**
(DOCX)

**S1 Fig.  Expression analysis of hsa-mir-183, hsa-mir-133b, and hsa-mir-145 in HNSCC.**
(DOCX)

**S2 Fig.  Protein interaction network of TP53 protein.**
(DOCX)

**S3 Fig.  ConSurf analysis of TP53's evolutionary conservation.**
(DOCX)

**S4 Fig.  This figure highlights the phosphorylation modifications of TP53, which may influence its activity and stability.**
(DOCX)

**S5 Fig.  (A) The surface of the TP53 pocket (PDB ID: 2ac0), computed by CASTp, with the pocket region indicated in red.** (B) The P53 protein dimer and DNA substrate, colored in yellow and magenta, respectively. The R273C, Q136P, Q136H, and R280G mutations are part of the active site.
(DOCX)

## Acknowledgments

We would like to express our sincere gratitude to Prof. Mohd Firdaus Raih and Dr. Edison Eukun Sage for their valuable contributions to this study. Special thanks are extended to the NUBRI team for their continuous technical support and guidance throughout the research process. We are also deeply grateful to Universiti Kebangsaan Malaysia and National Uinversity Biomedical Research Institute, National University-Sudan for providing the essential infrastructure and tools necessary to carry out this study. Finally, we thank Prof. Qurashi M. Ali for their insightful feedback and collaborative support.

## Author contributions

**Conceptualization:** Sofia B. Mohamed.

**Formal analysis:** Ashraf Attia Mahmoud, Sabah A. E. Ibrahim, Osama Mohamed.

**Methodology:** Ashraf Attia Mahmoud, Edison Eukun Sage, Omnia H. Suliman.

**Project administration:** Sofia B. Mohamed.

**Supervision:** Samar Abdelrazeg, Sofia B. Mohamed.

**Validation:** Mohd Firdaus Raih, Sofia B. Mohamed.

**Writing – original draft:** Ashraf Attia Mahmoud, Samar Abdelrazeg, Sofia B. Mohamed.

**Writing – review & editing:** Mohd Firdaus Raih, Qurashi M. Ali, Sofia B. Mohamed.

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
