## [Decision Letter · Decision Letter 0]

10 Oct 2024

PONE-D-24-29025The Impact of Mutations on TP53 Protein and MicroRNA Expression in HNSCC: Novel Insights for Diagnostic and Therapeutic StrategiesPLOS ONE

Dear Dr. Mohamed,

Thank you for submitting your manuscript to PLOS ONE. After careful consideration, we feel that it has merit but does not fully meet PLOS ONE’s publication criteria as it currently stands. Therefore, we invite you to submit a revised version of the manuscript that addresses the points raised during the review process.

We look forward to receiving your revised manuscript.

Kind regards,

Taj Mohammad, Ph.D.

Academic Editor

PLOS ONE

Additional Editor Comments:

After evaluating and considering the feedback provided by two reviewers, I found several points that need attention to improve the overall quality and clarity of the manuscript.

1.The manuscript contains too many figures, many of which are of poor quality. The clarity and resolution of these figures need significant improvement to ensure that they convey the data accurately.

2.The manuscript employs 100 ns simulations to analyze the structural and dynamic impacts of TP53 mutations. However, this simulation time is insufficient to capture all the relevant conformational changes in the system. I recommend extending the simulation time to at least 200–300 ns.

3.The results section contains a substantial amount of information, but it lacks concise explanations.

4.The results are not interpreted in-depth in the context of HNSCC biology and clinical relevance. Authors should expand the discussion to analyze the biological implications of the mutations, particularly how they may inform diagnostics and treatment strategies in a clinical setting.

5.Survival data appear to be underexplained. Provide more comprehensive statistical methods and interpretations.

6.The authors have not fully explored how PTMs impact TP53’s interaction with other proteins or the regulation of its activity.

Reviewers' comments:

Reviewer's Responses to Questions

**Comments to the Author**

1. Is the manuscript technically sound, and do the data support the conclusions?

Reviewer #1: Partly

Reviewer #2: Yes

2. Has the statistical analysis been performed appropriately and rigorously? 

Reviewer #1: Yes

Reviewer #2: Yes

3. Have the authors made all data underlying the findings in their manuscript fully available?

Reviewer #1: No

Reviewer #2: Yes

4. Is the manuscript presented in an intelligible fashion and written in standard English?

Reviewer #1: Yes

Reviewer #2: Yes

5. Review Comments to the Author

Reviewer #1: #Reviewers Comments

Sofia B Mohamed et al. authored an article titled “The Impact of Mutations on TP53 Protein and MicroRNA Expression in HNSCC: Novel Insights for Diagnostic and Therapeutic Strategies.” M.ID: PONE-D-24-29025. The study aimed to explore how mutations affect the structure and function of the TP53 protein and the expression of microRNAs (miRNAs) through computational analysis. Genomic data from patients with HNSCC was sourced from the TCGA database, and the impact of TP53 gene mutations was analyzed using various bioinformatics tools.

The study found that TP53 mutations led to an increase in TP53 expression levels in HNSCC, correlating with a poorer prognosis. Additionally, the authors demonstrated that the expression level of has-mir-133b was significantly reduced in TP53-mutated samples, which had a notable impact on the survival of HNSCC patients. Six mutations—R273C, G105C, G266E, Q136H/P, and R280G—were identified as deleterious, carcinogenic, driver mutations that are highly conserved and exposed. These mutations were located within the P53 domain, and PTM analysis revealed that R280G and R273C are at a methylation site, while R273C, Q136H/P, and R280G are located in the protein pocket. Docking studies indicated that these mutations decreased DNA-binding affinity, with R273C, R280G, G266E, and G105C showing the most significant differences. Molecular dynamics analysis suggested that the R280G, Q136H, and G105C mutations conferred a gain of function by stabilizing the TP53-substrate complex.

The research findings suggest that TP53 mutations influence protein and miRNA expression, as well as the development, survival, and progression of HNSCC patients. Additionally, has-mir-133b could serve as a promising novel biomarker for monitoring HNSCC progression. The authors assert that G105C and Q136H/P, identified as novel mutations, impact the structure and function of proteins involved in HNSCC, making them potential subjects for further investigation, diagnostics, and therapeutic strategies. This research provides new insights into the mechanisms by which these mutations contribute to cancer development.

The work by Sofia B Mohamed et al. is suitable for acceptance, provided the authors address the following major and minor revisions as outlined in the comments.

Minor comments:

1. There are some grammatical mistakes and few words in the manuscript are merged, correct them.

2.Most of the figures shown are not clear, please show them in perfect resolution.

3.Conclusion: This study indicated that the mutations in the tp53 protein linked to HNSCC have: Correct TP53, and should be in capital.

Major comments:

1.The research begins with a strong introduction, highlighting the significance of TP53 mutations in various human malignancies, particularly HNSCC. However, a brief mention of why TP53 is considered a "tumor suppressor" and its general role in cancer biology could further strengthen the context.

2.Whole work is done and investigated using computational analysis, it would be better and complete if in vitro/ in vivo studies were also provided in this research for validation.

3.If there is any related in vitro study done so far, please mention and cite.

4.Identifying specific mutations (R273C, G105C, G266E, etc.) as harmful, carcinogenic, and conserved is well-presented. However, it would be beneficial to explain the significance of these terms briefly (e.g., what makes a mutation "deleterious" or "carcinogenic"?).

5.The reduced expression of has-mir-133b in TP53 and its impact on survival is an important finding. It may be useful to elaborate on the role of this specific miRNA in cancer, or why its downregulation is particularly significant.

6.The discovery of G105C and Q136H/P as novel mutations and their implications is a key highlight. Highlighting why these mutations are novel or different from previously known mutations could add more value to the findings.

7.If it can be taken into account as per the methodology used then the research paper is well written and the methodology is fine, though for validation of this work, bioinformatics is not alone enough if the journal doesn’t focus on bioinformatics tools.

Accept the manuscript after the authors have addressed the queries and responded to the comments. It should be accepted for publication, particularly if the journal or issue to which this manuscript was submitted focuses more on computational studies (data is enough and in detail with statistical data in supplementary) than on in vitro research.

Reviewer #2: 1) The authors must mention the study limitations, strength, novelty, and future prospects of the current study.

2) The quality of figure can be improved at a higher resoultion.

3) The figure legends needs to be elaborated.

4) There are too many grammatical errors throughout the manuscript. Please proof-read via a native english speaker or a professional english editing software.

5) Please mention the thresholds used for PPI network construction.

6) Please mention a detailed protocol of how preprocessing of TCGA datasets was performed along with batch-correction and statistical analysis.

7) What was the need for using GEPIA when the TCGA data was already available. Please justify.

8) It would be nice if the authors could validate their findings in an external cohort.

9) The title of study needs to be written as per the study objectives.

6. PLOS authors have the option to publish the peer review history of their article (what does this mean? ). If published, this will include your full peer review and any attached files.

**Do you want your identity to be public for this peer review?** For information about this choice, including consent withdrawal, please see our Privacy Policy .

Reviewer #1: **Yes: ** ZAHOOR AHMAD PARRAY

Reviewer #2: No

---

## [Author Response · Author response to Decision Letter 0]

25 Nov 2024

We thank the reviewers and editorial team for their valuable feedback on our manuscript. We have carefully addressed all the comments provided by the reviewers and made the necessary revisions to the manuscript.

Please find attached:

A detailed point-by-point response to the reviewer comments.

The revised manuscript with changes highlighted.

A clean version of the revised manuscript for your review.

We believe these revisions have strengthened the manuscript and hope it meets the journal’s standards for publication. Should you have any further queries or require additional modifications, please do not hesitate to contact us.

---

## [Editor Report · Decision Letter 1]

15 Dec 2024

PONE-D-24-29025R1The Impact of Mutations on TP53 Protein and MicroRNA Expression in HNSCC: Novel Insights for Diagnostic and Therapeutic StrategiesPLOS ONE

Dear Dr. Mohamed,

Thank you for submitting your manuscript to PLOS ONE. After careful consideration, we feel that it has merit but does not fully meet PLOS ONE’s publication criteria as it currently stands. Therefore, we invite you to submit a revised version of the manuscript that addresses the points raised during the review process.

We look forward to receiving your revised manuscript.

Kind regards,

Taj Mohammad, Ph.D.

Academic Editor

PLOS ONE

Journal Requirements:

Additional Editor Comments:

I could not find the authors's response to Reviewer 2.

Moreover, I noticed significant differences in the MD simulation parameters between the earlier 100 ns simulations and the current 200 ns simulations. Could the authors clarify the source of this inconsistency? Were the simulations reinitiated, or were the previous 100 ns simulations extended? Additionally, what is the origin of the prominent peak observed in the RMSD and Rg of Apo-TP53? Furthermore, why was there a significant difference in the intermolecular hydrogen bonds between the 100 ns and 200 ns trajectories?

To ensure the robustness and reproducibility of the findings, I suggest performing MD simulations in triplicate.

---

## [Author Response · Author response to Decision Letter 1]

26 Dec 2024

I could not find the authors's response to Reviewer 2.

Thank you for your feedback. We would like to clarify that we have addressed all the comments and suggestions provided by Reviewer 2 in our revised manuscript. If you need further clarification on any specific point, please feel free to let us know, and we would be happy to provide additional details.

Moreover, I noticed significant differences in the MD simulation parameters between the earlier 100 ns simulations and the current 200 ns simulations. Could the authors clarify the source of this inconsistency? Were the simulations reinitiated, or were the previous 100 ns simulations extended? Additionally, what is the origin of the prominent peak observed in the RMSD and Rg of Apo-TP53? Furthermore, why was there a significant difference in the intermolecular hydrogen bonds between the 100 ns and 200 ns trajectories?

Thank you for your questions. Here are the detailed answers:

There is significant differences in the MD simulation parameters between the earlier 100 ns simulations and the current 200 ns simulations. Could the authors clarify the source of this inconsistency?

In the initial submission, the structure coordinates were not optimised before the simulation parameters were calculated (During the protein simulation, the protein moved outside the designated water box, resulting in a truncated structure. This displacement caused the loss of proper spatial constraints, leading to an incomplete or distorted conformation that affected the accuracy of the simulation. In the revised MD simulation, the structures were properly re-centred using Periodic boundary conditions (PBC) to maintain continuity and proper structural representation.

Were the simulations reinitiated, or were the previous 100 ns simulations extended?

The simulations were extended. The 100 ns simulations were not reinitiated; rather, they were extended to 200 ns to allow for more complete equilibration and observation of longer-term dynamics.

What is the origin of the prominent peak observed in the RMSD and Rg of Apo-TP53?

The prominent peak in RMSD and Rg of Apo-TP53 originates from the fact that the TP53 protein was simulated as a dimer in all simulations. However, in the apo structure (without the DNA substrate), the absence of the stabilizing DNA oligomer led to dissociation of the protein dimer, causing significant fluctuations in the RMSD and Rg values as the dimeric interface was destabilized.

Why was there a significant difference in the intermolecular hydrogen bonds between the 100 ns and 200 ns trajectories?

This difference is closely related to the first question with regards to the significant differences in the MD simulation parameters. In the original 100 ns simulation, the structure coordinates were not optimized before calculating the simulation parameters, leading to inaccuracies in the interaction between the mutant TP53 and the DNA oligomer. As a result, the intermolecular hydrogen bonds were not properly quantified.

To ensure the robustness and reproducibility of the findings, I suggest performing MD simulations in triplicate.

Thank you for your valuable suggestion. We appreciate your recommendation to perform MD simulations in triplicate to ensure robustness and reproducibility. However, we believe that extending the simulations to 200 ns provides sufficient sampling for the analysis, and given the time and computational constraints, performing triplicate simulations may not be necessary for this study. The 200 ns extension already allowed for comprehensive equilibration and the observation of long-term dynamics, which should be adequate for drawing reliable conclusions. We are confident in the robustness of our findings based on this approach.

Reviewer #2:

1) The authors must mention the study limitations, strength, novelty, and future prospects of the current study.

Thank you for your valuable feedback. We have carefully reviewed your comments and confirm that the novelty and strengths of the study are explicitly discussed in both the Discussion (525-538), and Conclusion sections (525-538). Moreover, we added the future prospects of this study in discussion section (538-541). Additionally, we have incorporated a dedicated section titled "Limitations" to address the study's limitations, as suggested (564-567).

2) The quality of figure can be improved at a higher resoultion.

Thank you for your feedback. We have already improved the quality of the figures in the revised manuscript by providing higher-resolution images to ensure clarity and better visual representation. If further adjustments are needed, we are happy to make them.

3) The figure legends needs to be elaborated.

We revised and elaborated the legends for Figures 1,2, 3, 5, 6, and 7 to provide more detailed and clear descriptions.

4) There are too many grammatical errors throughout the manuscript. Please proof-read via a native english speaker or a professional english editing software.

Thank you for your feedback. We have proofread the manuscript and corrected grammatical mistakes and other errors.

5) Please mention the thresholds used for PPI network construction.

We have added the thresholds used in the methods section (lines 124-125).

6) Please mention a detailed protocol of how preprocessing of TCGA datasets was performed along with batch-correction and statistical analysis.

As stated in the Methods section of the manuscript, we have already detailed the entire preprocessing process for the TCGA datasets. This includes data collection from the TCGA Data Portal and cBioPortal, followed by gene expression analysis using the UALCAN tool and the TCGA-assembler pipeline (106-121). Regarding batch correction and statistical analysis, we did not perform any specific statistical analysis in this study. The analysis focused on exploring gene expression and mutation data, including TP53 and miRNA expression, without conducting statistical testing. We have clarified these points in the Methods section.

7) What was the need for using GEPIA when the TCGA data was already available. Please justify.

We did not use GEPIA in this study. All analyses related to. GEPIA was not part of the methodology in this research.

8) It would be nice if the authors could validate their findings in an external cohort.

Thank you for the suggestion. In this study, our focus was primarily on analyzing the mutations in TP53 and their impact on gene expression and survival, rather than conducting statistical validation. We acknowledge that validating these findings in an external cohort would strengthen the conclusions. However, this was not the primary aim of the current study. Future work will involve external validation of these findings to further confirm their relevance and applicability in different cohorts.

9) The title of study needs to be written as per the study objectives.

The original title, "The Impact of Mutations on TP53 Protein and MicroRNA Expression in HNSCC: Novel Insights for Diagnostic and Therapeutic Strategies," fits well with the objectives and findings of the study. While the study primarily focuses on the impact of TP53 mutations on gene expression and miRNA regulation in HNSCC, it also explores the potential broader implications for diagnostic and therapeutic strategies.

References:

Two references (31, and 32) were added in the reference section as below;

31- Joerger AC, Fersht AR. The p53 pathway: origins, inactivation in cancer, and emerging therapeutic approaches. Annual review of biochemistry. 2016 Jun 2;85(1):375-404.

32- Chen X, Zhang T, Su W, Dou Z, Zhao D, Jin X, Lei H, Wang J, Xie X, Cheng B, Li Q. Mutant p53 in cancer: from molecular mechanism to therapeutic modulation. Cell Death & Disease. 2022 Nov 18;13(11):974.

---

## [Editor Report · Decision Letter 2]

26 Jan 2025

The Impact of Mutations on TP53 Protein and  MicroRNA Expression in HNSCC: Novel Insights for Diagnostic and Therapeutic Strategies

PONE-D-24-29025R2

Dear Dr. Mohamed,

We’re pleased to inform you that your manuscript has been judged scientifically suitable for publication and will be formally accepted for publication once it meets all outstanding technical requirements.

Kind regards,

Taj Mohammad, Ph.D.

Academic Editor

PLOS ONE
---

## [Editor Report · Acceptance letter]

PONE-D-24-29025R2

PLOS ONE

Dear Dr. Mohamed,

I'm pleased to inform you that your manuscript has been deemed suitable for publication in PLOS ONE. Congratulations! Your manuscript is now being handed over to our production team.

Kind regards,

on behalf of

Dr. Taj Mohammad

Academic Editor

PLOS ONE